# A Simple Method to Quantify Outward Leakage of Medical Face Masks and Barrier Face Coverings: Implication for the Overall Filtration Efficiency

**DOI:** 10.3390/ijerph19063548

**Published:** 2022-03-16

**Authors:** Silvia Chiera, Alessandro Cristoforetti, Luca Benedetti, Giandomenico Nollo, Luca Borro, Lorenzo Mazzei, Francesco Tessarolo

**Affiliations:** 1Department of Industrial Engineering, University of Trento, 38123 Trento, Italy; silvia.chiera@unitn.it (S.C.); alessandro.cristoforetti@unitn.it (A.C.); luca.benedetti@unitn.it (L.B.); giandomenico.nollo@unitn.it (G.N.); 23DLab, Imaging Department, Bambino Gesù Children’s Hospital, 00165 Rome, Italy; luca.borro@opbg.net; 3Ergon Research, 50127 Florence, Italy; lorenzo.mazzei@ergonresearch.it; 4Healthcare Research and Innovation Program (IRCS-FBK-PAT), Bruno Kessler Foundation, 38123 Trento, Italy

**Keywords:** COVID-19, SARS-CoV-2, medical face masks, surgical face masks barrier face coverings, community masks, air leak, filtration efficiency, fit testing, breathability

## Abstract

Face masking proved essential to reduce transmission of COVID-19 and other respiratory infections in indoor environments, but standards and literature do not provide simple quantitative methods for quantifying air leakage at the face seal. This study reports an original method to quantify outward leakage and how wearing style impacts on leaks and filtration efficiency. The amount of air leakage was evaluated on four medical masks and four barrier face coverings, exploiting a theoretical model and an instrumented dummy head in a range of airflows between 30 and 160 L/min. The fraction of air leaking at the face seal of the medical masks and barrier face coverings ranged from 43% to 95% of exhaled air at 30 L/min and reduced to 10–85% at 160 L/min. Filter breathability was the main driver affecting both leak fraction and total filtration efficiency that varied from 5% to 53% and from 15% to 84% at 30 and 160 L/min, respectively. Minor changes were related to wearing style, supporting indications on the correct mask use. The fraction of air leaking from medical masks and barrier face coverings during exhalation is relevant and varies according to design and wearing style. The use of highly breathable filter materials reduces air leaks and improve total filtration efficiency.

## 1. Introduction

In early 2020, the World Health Organization declared the pandemic phase of the new SARS-CoV-2 virus [1], an RNA virus belonging to the coronavirus family capable of spreading worldwide causing over several million deaths. The virus was first isolated in Wuhan, China, in December 2019 and it was associated with severe cases of interstitial pneumonia [2]. The disease resulting from infection by the SARS-CoV-2 virus was called COVID-19 syndrome. The transmission of SARS-CoV-2 mainly occurs through the airways by the saliva droplets emission and by the aerosols generated by respiration [3,4], while a minor role is associated with contact with contaminated surfaces [5,6,7,8]. According to the main route of transmission, the use of face masks to cover mouth and nose proved to be among the most effective tools for limiting the spread of SARS-CoV-2 [9] resulting in a significant decrease of several infectious respiratory diseases [10,11].

Recent studies showed that the correct and widespread use of face masks among the population contributes significantly to limiting the pandemic [4,12,13], other than having positive health effects by reducing particulate matter exposure [14,15]. These findings support, up to the current date, the enforcement of specific recommendations and regulations about the use of face masks for the population in many countries [16,17,18]. As a consequence, the world’s population have had to profoundly change their lifestyle and social behaviours since February 2020, including, among others, the wearing of face masks, especially in indoor environments.

The consequent and sudden increase in demand for protective devices resulted in a failure of the supply chain, with shortages of personal protective equipment [19], particularly of face masks [20]. This contingency generated the need for starting local production of face masks, exploring alternative systems and materials for protecting the nose and the mouth from potentially SARS-CoV-2 contaminated droplets and aerosol, favouring a capillary diffusion and wide availability of new products among the population [21]. New terms like “homemade masks” [22,23], “barrier face coverings” [24] or “community masks” [25,26] (hereafter referred as CMs), were adopted to distinguish between the so called “medical” or “surgical” face masks (hereafter referred as SMs), which perform according to requirements set by standards, and those products and hand-made solutions, whose performance is either substandard or not assessed by the manufacturer.

Although the World Health Organization published an interim guidance [27] underlying that breathability, filtration efficiency, and face fit are essential characteristics to be considered also for non-medical masks, at present the performance of CMs is not regulated by quality standards, and only a discretionary guide to minimal requirements has been made available by CEN [25]. ASTM recently provided a framework for the specifications of barrier face covering performance [24], however the document recognizes that there are currently no specific accepted techniques to measure outward leakage from CMs.

In contrast, the available standards for SMs [28] specify requirements for filtration efficiency and for breathability. However, the methods and equipment involved by the standards [28,29] are designed to test characteristics of the filtering materials and do not evaluate the mask design and its fitting on the user’s face. Therefore, no requirements are set concerning the amount of air that can flow at the mask–face interface without being filtered.

Testing methods devoted to quantifying the total filtration efficiency, taking into account both material filtration and leakage quantification, are currently applied only to filtering face piece respirators (FPs) and are specified in related standards [30]. Their application is complex and requires both the use of dedicated equipment and the involvement of human test subjects, limiting its applicability only to large standardization test centres.

The exhaled air leaking from the face mask worn by an infected person can play a critical role in virus transmission as several studies evidenced the importance of its evaluation [31,32,33,34,35,36,37,38]. Numerous experiments have been conducted to evaluate the total filtration efficiency of SMs and CMs with the aim of understanding their capability to contain viral spread [13,34,39,40]. These studies mainly performed qualitative tests and were able to demonstrate the physics of droplets and aerosols, their aerodynamic capabilities, and how these respiratory fluids can spread within different environments. These studies reported only qualitative data, showing leak flow preferential direction, droplet projection distance, or aerosol density distribution, without providing numbers on the total filtration efficiency. Only recently, Lindsley and co-workers reported data about the mass fraction of respiratory aerosol that is blocked by cloth masks by using a complex apparatus composed of realistic head form, a large aerosol chamber, and a multistage aerosol impactor [41].

As pointed out by a randomized clinical trial [42] and by recent computational studies [38], leakage is further dependent on mask position on the user face, thus being impacted by user wearing style and compliance to manufacturer’s indication for use.

To properly quantify the total outward filtration efficiency of SMs and CMs, two pathways should be considered for the exhaled air: leakage through the face seal, and flow through the filter medium [33]. Pre-pandemic data indicated that a relevant amount of particles can pass through the face seal, suggesting that, in mask design, priority should be given to establishing the optimal fit for minimizing face seal leakage [33]. It is therefore urgent to complement the measure of filtration efficiency of the filtering material with the quantification of the fraction of exhaled air leaking at the face seal.

For these reasons, the aim of this study was threefold: first, to define a simple method to quantify the fraction of exhaled air leaking from SMs and CMs in relevant conditions of use; second, to evaluate the impact of different wearing styles on the amount of air leakage; third, to produce quantitative data of total outward filtration efficiency for SMs and CMs, combining the bacterial filtration efficiency of the filtering materials with the leaking fraction information.

## 2. Materials and Methods

The experimental approach of this study consisted in emulating the exhalation of air through a face mask worn by a person during normal breathing, exploiting an instrumented head manikin. Experimental flow data were analyzed and interpreted based on the theoretical framework describing the main paths followed by the exhaled air.

### 2.1. Theoretical Model

In real conditions, the face mask, covering mouth and nose, creates a certain resistance to the exhaled airflow, which depends primarily on the mask materials and the whole mask design, including the face fitting and the overall breathability. The total airflow exhaled by the mouth *Q_I tot_* (where the subscript “*I*” refers to real conditions) splits into two components, *Q_I mask_* passing through the mask filter, and *Q_I leak_* leaking at the face seal:(1)$$Q_{I\;tot}=Q_{I\;mask}+Q_{I\;leak}$$

Consequently, airflow resistance can be subdivided in two components, one associated to the airflow passing through the mask material (*R_mask_*), and a second component associated to the airflow escaping through the critical constriction at the mask boundaries (*R_leak_*). Considering that no face mask available on the market has a null airflow resistance (ideal scenario), a differential pressure Δ*P_I_* is present between the inside of the mask and the external environment, which is the common driver for both *Q_I mask_* and *Q_I leak_*. *Q_I mask_* is determined by the resistance *R_mask_* of the mask filter. According to Darcy’s law [43], the volumetric flow rate of a fluid with a viscosity µ through the porous medium having a cross-sectional area *A*, a thickness *L*, and a permeability *k*, is proportional to the pressure drop applied across the porous medium [36]. This allows modelling the flux through the mask material as:(2)$$\Delta P_{I}=Q_{I\;mask}\cdot R_{mask}\;$$
where the flow resistance *R_mask_* can vary according to the characteristics of the filter as follows:(3)$$R_{mask}=µL/kA$$

Unfortunately, *Q_I leak_* cannot be simply modelled or calculated, being related to several factors including flow velocity and size and shape of the openings at the face seal interface.

For the sake of measuring *Q_I leak_* by the experimental procedure detailed below, we also considered the flux model for the ideal situation, where no air leak is present (perfect mask fit to the face). In this case, the total airflow is represented exclusively by the flow passing through the mask *Q_II mask_*, which can be related to the differential pressure $\Delta P_{II}$ measured in ideal conditions by an equation similar to Equation (2) (where the subscript “II” refers to ideal conditions):(4)$$\Delta P_{II}=Q_{II\;mask}\cdot R_{mask}$$

### 2.2. Experimental Set Up

The experimental setup was based on a polylactic acid dummy head 3D-printed according to the dimensional characteristics of the medium sized head, as specified by the standard ISO 16900-5. The surface was finished with sandpaper and epoxy resin to remove porosity. The dummy head was then instrumented with a piping system to generate a controlled airflow through the mouth region, as specified in part 8.9 of the EN 149:2009 standard [30]. Briefly, the mouth opening consisted in the open end of a 42 mm diameter tube, devoted to simulating air inhalation (not used in this study), with an inner concentric tube of 28 mm in diameter for air exhalation. A third smaller tube (6 mm in diameter) was also present at the centre of the mouth opening and was used to sample the pressure at this point by using a differential manometer. A compressed air supply mimicked exhalation generating a constant airflow through the 28 mm tube, measured by a dedicated flowmeter. A circular grid was fixed around the boundaries of the mouth opening to allow a homogeneous spread of the flow outside the mouth, preventing direct contact between the outlet and the mask surface even when the mask was tightly pressed on the dummy head. A comprehensive view of the experimental set up is presented in Figure 1. An air flow rate between 30 and 160 L/min was generated during the study, covering the airflow values specified for testing face respirators according to the standards [30] and the typical flow rate generated during speech [39]. A digital flow sensor having a resolution of 1 L/min and a 3% (+1 LSD) accuracy (Digital Flow Switch PFM7, SMC Corporation, Tokyo, Japan) was used to measure the airflow rate exiting the dummy head in steady-state conditions. The flow meter provided volumetric flow measurements corrected to standard conditions (101.3 kPa, 20 °C, 65% RH). A differential manometer (Fluke 992, Fluke Corp., Everett, WA, USA), having a resolution of 1 Pa and an accuracy of 1% (+1 LSD) was used to measure the Δ*P* occurring between the dummy head mouth opening and the external environment.

To study and quantify the mask leaks in relation to the total exhaled airflow, the experiment was performed in two phases. The first phase (Phase *I*) mimicked the real situation where leaks are present. The tested mask was positioned on the dummy head following manufacturer’s instructions for use (IFU), covering both nose and mouth, hanging the ear laces or the rubber bands at their intended position, and conforming the mask border as much as possible to the dummy head profile by making some pressure with fingers. No extra means were applied to set the mask in place. Special attention was posed when adapting the nose piece, whenever present, with the aim of maximizing the mask seal. 

When the mask was properly positioned, a constant airflow (*Q_I tot_*) was generated at 30 L/min, and the corresponding value of differential pressure (Δ*P_I_*) was collected in a steady state condition (approximately after 5 s from reaching the expected airflow rate) (Figure 2a). Then the same procedure was repeated for different airflow rates covering the range from 30 to 160 L/min at intervals of 10 L/min, collecting the corresponding values of differential pressure (see Equation (2)).

The second phase (Phase *II*) of the experiment emulated the ideal situation of a perfect mask fit (*Q_leak_* = 0). The same procedure described in Phase I was repeated after sealing the mask boundaries on the dummy head using adhesive tape (paper masking tape, 25 mm width, Tesa Masking Economy, Tesa SE Hamburg, Germany). The masking tape was applied across the mask filter perimeter and the surface of the dummy head. Special care was paid to cover only the peripheral welded areas of the mask filter that do not contribute to the filter airflow. Manual pressure was made on the tape to guarantee no air leaking was present through the sealed areas. A visual inspection of the tape adherence to the mask boundaries and to the dummy head was performed at the end of each Phase *II* experiment. This procedure allowed us to assume no leakage was present (Figure 2b). A second set of differential pressure measurements, indicated as Δ*P_II_* (see Equation (4)), was then collected in this experimental condition, with airflow rates (*Q_II mask_*) ranging from 30 to 160 L/min at intervals of 10 L/min.

In the first phase, the pressure increase (Δ*P_I_*) was determined by a combination of the resistance of the mask filter (*R_mask_*) and the resistance of the air leaking at the face seal constrictions. In the second phase, the whole airflow (*Q_II mask_*) was forced to pass through the mask material, and the pressure increase (Δ*P_II_*) depended solely on the resistance of the mask (*R_mask_*). Based on the previously explained mathematical model, the integrated analysis of the differential pressure values measured in the two experimental conditions allowed the calculation of the exhaled air flow repartition between the mask filter (*F_mask_*) and leakage at the face seal (*F_leak_*), as detailed in the section below.

### 2.3. Data Analysis

In the experimental conditions of Phase *I*, the measurements of differential pressure Δ*P_I_* and of total airflow *Q_I tot_* did not allow to calculate the values of *Q_I mask_*, and *Q_I leak_*, since *R_mask_* is unknown. However, the measurements of Δ*P_II_* and *Q_II mask_* obtained in the experimental conditions of Phase *II* allowed us to determine *R_mask_* every 10 L/min in the airflow range from 30 L/min to 160 L/min by using Equation (4).

The experimental values of *R_mask_* were then used to build an analytical profile of *R_mask_* as a function of Δ*P_II_*, by fitting a first-order polynomial curve on the measured *R_mask_* data. Assuming that the mask filter behaviour did not substantially change between Phase *I* and Phase *II* configurations for equivalent values of differential pressure, preserving the mask resistance characteristics, the analytical profile of *R_mask_* was used to calculate the values of *R_mask_* for the Δ*P_I_* values measured in the experimental conditions of Phase *I*. These resistance values were then used in Equation (2) to compute the values of *Q_I mask_* corresponding to Δ*P_I_*. Finally, the corresponding *Q_I leak_* values were obtained considering flow mass conservations expressed in Equation (1). 

For each value of total exhaled airflow in the 30–160 L/min range, the percentage of airflow leaking at the face seal was calculated according to the following equation:(5)$$F_{leak\;}\left(\%\right)=100\frac{Q_{I\;leak}}{Q_{I\;tot}}.$$

Similarly, the complementary percentage of airflow passing through the mask filtration material was defined according to:(6)$$F_{mask}\left(\%\right)=100\frac{Q_{I\;mask}}{Q_{I\;tot}}.$$

Having obtained *F_leak_* and *F_mask_*, the total mask outward filtration efficiency (*TFE*) in the 30–160 L/min range was calculated using the following equation:(7)$$TFE\left(\%\right)=F_{mask}\cdot BFE/100$$
where BFE was the bacterial filtration efficiency of the mask filter, measured according to the method specified in Annex B of standard EN 14683:2019 [28].

### 2.4. Facemasks and Respirators Tested in the Study

The experimental protocol was applied to eight different face masks representative of the most common types available on the market. In order to analyze how variations in design, manufacturing materials, and filtering efficiency affected flow repartition, the mask group included both community masks (CM) and surgical face masks (SM) of type I, II, and IIR. Two commercially available filtering face piece (FP) respirator models were also added in the study, as reference for minimal airflow leakage as expected by their face fit design. All the masks and respirators were tested covering both mouth and nose of the dummy head, following the instructions for use (IFU), as shown in Figure 3.

Specifications of masks and respirators are reported in Table 1, including the values of breathability (DP, Pa/cm^2^) and bacterial filtration efficiency (BFE, %), both checked independently at our laboratory according to the methods of Annex B (bacterial filtration efficiency, BFE) and Annex C (breathability, DP) of standard EN 14683:2019 and using the equipment previously presented [21]. These tests were applied as well on the filtering materials of the two FP models to allow a *TFE* comparison with the SM and CM mask models. In addition, mask category and type as claimed by the manufacturer, and some essential information about the filtering material, the nose piece and ear loops design are listed.

In addition to the IFU position, alternative mask wearing styles were investigated to understand how they could impact outward flow repartition and in particular the leakage fraction. To imitate the most common ways of wearing a mask among people, three different positions have been identified and tested on the eight face masks: (a) mask lowered, with the upper band of the mask on nose tip, (b) mask with laces crossed at the ears, and (c) mask with laces gathered at the nape. An ear-saver was used in the last case to keep the mask laces always at a predefined distance of 8.5 cm during the test. The different wearing styles assessed in this study are shown in Figure 4 for a typical medical face mask. The face respirators were not tested under different wearing conditions.

### 2.5. Statistical Analysis

Experiments were performed in quintuplicate for each mask model, exhaled flow rate, and wearing style. The average over the five replicates was calculated, and the standard deviation of the repeated measurements was considered as an indicator of repeatability. The instrumental uncertainty associated to *F_leak_, F_mask_* and *TFE* values was obtained by propagating the instrumental errors on differential pressure and airflow rate measurements. The overall uncertainty was obtained considering the highest between repeatability and instrumental uncertainty at each experimental condition (i.e., combination of mask model, exhaled flow rate, wearing style).

For each mask model, values of *F_leak_* obtained under IFU wearing and under alternative wearing styles were compared using the Friedman non-parametric statistical test. Post hoc correction was applied according to Dunn’s multiple comparison test. A *p*-value < 0.05 was considered statistically significant. Statistical analysis was performed using Prism 5 statistical software (GraphPad Software, San Diego, CA, USA).

## 3. Results

### 3.1. Mask Resistance Features

The experimental values of the mask filter resistance (*R_mask_*) as a function of the differential pressure (Δ*P_II_*) are presented in Figure 5. The first-order polynomial best fit of the experimental dataset for each mask model is also reported. Data inspection revealed that mask resistance values were different among the tested models and minor variations were present for the same model within the explored range of differential pressure. CM-a showed the highest value of resistance in comparison to all the others. All tested models, except CM-b, showed a minor linear increase of resistance with the increase of the differential pressure. This could be ascribed to a compression of the filter at higher differential pressure. Conversely, CM-b showed an opposite trend which could be associated to the fiber structure of the filter, made of knitted fabric. This fiber arrangement allows for high elasticity of the fabric, that typically expands its surface under an increasing differential pressure. The increase in filter surface and porosity could possibly be the reason behind this different resistance behavior. An additional possible reason for the lowering of CM-b mask resistance at higher values of differential pressure could be related to a “lift off” effect in the area close to the nose of the dummy head, due to the absence of a nose piece and the elasticity of the mask boundary.

The overall data trend confirmed that a first-order polynomial curve was a suitable analytical model for representing the changes of the mask resistance as a function of the differential pressure applied at the filter surface.

### 3.2. Leak Quantification according to Mask Type

Flow repartition between the mask surface and the leaks for face masks worn according to IFU at different airflow rates is shown in Figure 6. Experimental data clearly indicate that flow repartition was mask-specific and varied with total outflow. However, a common trend was present in all tested face mask models, consisting in a decreasing percentage of air leaks from lower to higher flow rates. These results were consistent also with those obtained with the two respirators models FP-a and FP-b. The values of *F_leak_* for each tested model are summarized in Table 2, reporting the leakage fraction at low (30 L/min), intermediate (90 L/min), and high (160 L/min) flow rates.

Leakage comparison among the tested masks evidenced that the SM models had a more uniform performance in term of leakage fraction across the whole flow-rate range. Except for SM-b, the other three tested SM models (SM-a, SM-c, SM-d) showed a leakage percentage around 86% at low flow rate, which decreased to almost 60% at high flow rates. Interestingly, the values of breathability reported in Table 1 for SM-a, SM-c, and SM-d showed similar characteristics, with DP values around 30 Pa/cm^2^, irrespective of the fact that these three surgical mask models included different types (Type I and Type IIR) and manufacturers. In contrast, the SM-b model, although claimed by the manufacturer as a Type I surgical mask, showed higher leakage fractions ranging from 72% to 91%. This difference could be associated with the poor breathability of this mask (DP = 77 Pa/cm^2^), casting doubt on whether this product actually met the specification of the standard to which it was supposedly manufactured and labelled (EN 14683:2019 breathability requirements prescribe DP < 40 Pa/cm^2^ for surgical mask type I and II and DP < 60 Pa/cm^2^ for surgical mask type IIR).

The flow repartition within the CM group was highly heterogeneous, showing leak fractions higher and lower than those of SMs. More specifically, CM-c performed best of all the SMs and CMs tested in this study, with a leak fraction in the 10–43% range. The relatively low leakage of this mask model, without a nose-piece able to provide a good nose fitting, could be related to the extremely good breathability (DP = 9,6 Pa/cm^2^) of the triple layer of spunbonded non-woven PP filtering material. Mask CM-d, also without a nose piece, presented a moderately worse performance especially at low flow rate, with a leakage fraction ranging from 29% to 74%, while showing a slightly better breathability. The remaining two CMs, CM-a and CM-b, presented a markedly different scenario, with a leakage ranging from 85% to 95% and from 66% to 86%, respectively. CM-a showed the highest values of leakage fraction among all the tested masks. Being manufactured with two layers of woven cotton fabric with high thread number, CM-a had very poor breathability (DP = 315 Pa/cm^2^), imposing most of the exhaled airflow to leak at mask–face interface, although a wire nose piece was in place. Conversely, the better breathability (DP = 56 Pa/cm^2^) associated to the CM-b filtering material made of two layers of knitted fabric, resulted in a lower leakage fraction (close to those of surgical masks), even if no nose piece was present. 

The comparison of SM and CM leak performance to that of face piece respirators (FP-a and FP-b) showed the superiority of the latter kind of PPE in limiting air leaks. As expected, the face piece respirator design and their tightening systems resulted in a better face fit and minimized air leaks, forcing most of the airflow through the filtering material. The FP-b mask, having a specific design, two rubber bands to keep the mask in place, and a nose piece formed by a metal wire and a foam strip, showed the best performance among all tested. Indeed, almost the totality of airflow passes through the mask (~90–95%), leaving just the 8–11% of outward airflow leaking at the face seal. This result is consistent with the requirements on total inward leaking for respirators labelled as FFP2 according to UNI EN 149 [30]. FP-a showed a moderately lower performance, with the outward leakage limited within the 18–33% range. While this is far from the inward leakage requirements set for FFP2 labelled respirators according to UNI EN 149 [30], we did not test inward leakage performance, which could differ from outward leakage performance due to lift-off effects.

### 3.3. Leakage Quantification according to Wearing Style

CM and SM models were further tested for evaluating possible leakage variations in relation to three wearing positions alternative to that reported in the IFU. Leakage fractions obtained using different wearing styles for each SM and CM model are shown in Figure 7. The values of *F_leak_* for each model tested according to the different wearing styles are summarized in Table 2, reporting the leakage fraction at low (30 L/min), intermediate (90 L/min), and high (160 L/min) flow rates. Data analysis showed that all the masks worn on the nose tip consistently produced the highest leakage values. In contrast, all the masks worn with laces gathered at the nape, except for CM-b, produced the lowest leakage fractions. Wearing styles referring to IFU and with laces crossed at the ears resulted in intermediate leakage values. 

The different behaviour of the CM-b mask could be ascribed to the manufacturing materials, using elastic components for the filtering area and the filter borders, allowing the mask to be accommodated on the face differently from the other more rigid masks.

According to statistical analysis, the values of *F_leak_* obtained for each mask, when worn according to IFU and alternative wearing styles, showed significant differences. As summarized in Table 3, the mask positioned on the tip of the nose always produced values of *F_leak_* higher than those obtained when worn according to IFU. This result was statistically significant in five out of the eight tested face mask models, indicating that this wearing style worsen the performance of the mask. Conversely, gathering the ear loops at the nape always resulted in lower values of *F_leak_*, again with a statistical significance in five out of eight mask models. Crossing the ear loops resulted either in higher or lower values of *F_leak_*, depending on the mask model, while statistically significant higher values of *F_leak_* were obtained only for two community masks.

### 3.4. Total Filtration Efficiency

*TFE* was an indicator of overall mask performance, combining leakage efficiency with the bacterial filtration efficiency (BFE) data listed in Table 1. The face respirators FP-a and FP-b indicated a BFE of 90 and >99%, respectively. The surgical masks complied with their standard, showing BFE values greater than 98%. The community masks showed a BFE range between 90 and 96%. *TFE* was calculated using our results on flow repartition according to Equation (7). The computed *TFE* of the tested SM, CM and FP models for a total exhaled airflow rate ranging from 30 to 160 L/min are presented in Figure 8. *TFE* values obtained at low (30 L/min), medium (90 L/min), and high (160 L/min) outward flowrates are summarized in Table 2.

The comparison of *TFE* curves (Figure 8a) with the corresponding data of *F_mask_* (Figure 8b) evidenced the driving role of *F_mask_* in determining *TFE*. As an example, mask CM-a with a BFE of 96% presents a *TFE* limited to 5–15%, due to its low breathability and high leakage fraction. Interestingly, community masks such as CM-c and CM-d showed higher *TFE* values than the surgical masks analyzed in this study, thanks to their higher breathability. It is interesting to note that, according to our results, a highly breathable community mask (e.g., CM-c) appears to outperform a commercially available face piece respirator (e.g., FP-a) in terms of *TFE* at medium and high flow rates. The *TFE* of the FP-b mask was never outperformed by the masks tested in this study, thanks to the optimal flow repartition of that mask model. BFE performance has, indeed, only a minor effect on TFE, bringing minor changes in the ranking of the tested masks (see, for example, the changes in ranking of SM-c and SM-d between the two graphs of Figure 8).

## 4. Discussion

During the pandemic period, the importance of wearing a mask to reduce the transmission rate of the disease has been largely demonstrated [4,13]. Evaluation of mask efficiency at filtering pathogen-containing aerosol and particles can be broken down into two aspects: the fraction of airflow which passes through the mask without escaping through leakages, and the capability of the mask material to filter such airflow [33]. The current European standard EN 14683:2019 for surgical face masks specifies the method for evaluating the filtration capability in terms of BFE and sets the minimal requirements for surgical masks to be labelled as type I or type II. Conversely, the same standard does not set any performance requirements regarding mask fit and outward leakage, and there are no measuring recommendations either. Nonetheless, the fraction of air leaking from the face seal plays a key role in the overall filtration efficiency and has an impact also on the whole mask breathability. In their recent work, Duncan and co-workers demonstrated the importance of the inward leakage in the protection efficiency of a mask and how this is influenced by the mask fit [44]. In the present study we measured the outward leakage on some typical community and surgical masks, providing a simple method for a quantitative estimation of the flow repartition at realistic flow rates. To understand the impact of user compliance to IFU on flow repartition, results were also obtained reproducing different styles of wearing the face mask. Unlike several previous studies that characterized the leakage in a qualitative way [31,35], this study provides a way of quantifying the leaks by making use of a realistic and standardized instrumented dummy head. The results we obtained indicated that a large fraction of the exhaled air was not subjected to mask filtration but leaked unfiltered at the face seal. The repartition between the amount of airflow passing through the mask filter and that leaking through the face seal was strictly linked to the specific mask design, including several factors such as mask size, filtering area size, nose piece type, lace tension, filtering material composition and folding, in combination with the filter breathability and the mask fit to the user face. A common trend for all the tested masks was present, showing that mask leakage decreased as the applied flow rate was increased. This behaviour could be ascribed to two different aspects, namely air-flow momentum and turbulence. As the flow rate is increased, the momentum of the air increases quadratically with the fluid velocity. This ultimately leads to a higher propensity of the fluid to permeate through the mask. In contrast, at low flow rates, the exhaled air molecules have low inertia and directional changes towards the face seal gaps are easier. Another possible role might be played by turbulence, which is indeed produced at high flow rate due to the strain rate of the flow at the exit of the mouth and to the impact with the mask. This results in a more chaotic flow condition that can hinder the flow through the lateral gaps. This explanation was not proved in the literature, but the effects were highlighted by Cappa et al. [45], who observed a dependence of the overall mask efficiency on the flow rate. Particularly when talking, the number of particles captured by the mask increased at a rate of about 5% per L/min [45], suggesting that less aerosol was leaked. Similar conclusions were drawn by Hariharan et al., who investigated the particle leakage with different respirators, measuring a reduction from 17–22% to 4% when the flow rate was increased from 10 L/min to 70 L/min [46]. Tang et al. studied the turbulent jet caused by coughing and suggested that, according to the typology of the mask worn, it is possible to thwart the trajectory of the jet, directing airflow towards the boundaries of masks with poor fit [47]. In agreement with that observation, the results of the present study indicated that mask breathability has a major consistent effect on leak performance, as masks with high breathability showed markedly less leakage. This is also in agreement with that recently reported by Duncan et al., suggesting that a high pressure drop may be partially responsible for mask leakage at least during exhalation [44]. Our work not only confirmed this hypothesis, but evidenced that high breathability can drastically limit the mask leakage, facilitating the passage of the airflow through the mask filter also when the face fit is not optimized. Conversely, a high mask resistance forces the airflow through the openings at the face seal, irrespective of the presence of nose pieces or of a correct wearing style. These findings are in line with the results of previous studies addressing the effect of superimposing two masks, showing an increased mask leaking due to the lower combined breathability [36]. 

Our data also indicate that mask design and the force generated by stretched ear-loops have a role on the adherence of the mask borders to the face, reducing air leaking. This was previously pointed out by Grinshpun and co-workers, concluding that the mask’s peripheral design should be carefully considered to establish a better fit and minimize the face seal leakage [33]. Several studies discussed the role of the mask sealing design, and the influence of the filter composition on the overall breathability [31,38,48,49]. Recently Schmitt et al. provided computational data showing that the protection efficiency of improperly fitted face piece respirators can drop below the protection level of properly worn surgical and community masks [38]. Consistently with that, our study quantitatively showed that the wearing style can have an impact on the leakage fraction, improving or worsening the flow repartition. Different wearing styles are indeed related to different face fittings that can result in changing the tension of the ear loops and the relative position of the mask over the nose. When the ear loops are gathered at the nape, the mask is slightly tighter to the cheeks, reducing the air leaks close to the nose and at the mouth sides. On the other hand, when the mask is worn on the nose tip, large openings form at the nose sides, favouring airflow leaking through that way. The leakage of the masks with laces crossed at the ears did not always result in a leak reduction, possibly because, while the openings around the nose are reduced by tighter laces, those at mouth sides are slightly increased due to the lace intersection. Finally, variations of the effective mask area exposed to the exhaled flow can be caused by filter folding (e.g., plied areas) and occlusions, such as in the area under the chin [31,50].

It is worth noting that, although the wearing style has an impact on the leak amount, this effect was secondary to the characteristics linked to the filtering material (namely, the breathability of the mask), which had the major role in leak management. This was evident in the data of CM-a, showing that changes to wearing style cannot remediate the poor leak performance of this mask, caused by the very low breathability of its filtering materials. 

Recently Lindsay and co-workers evidenced that none of the standard metrics such as filtration efficiency, breathability or airflow resistance, and manikin or human fit factors, are direct measurements of how effectively a mask blocks coughed and exhaled aerosols, and that none of them strongly correlated with source control performance [41]. The proposed *TFE* should address the need for a comprehensive description of mask efficiency, integrating both filter properties, breathability and fit into a single performance parameter. This concept could also be further integrated in more complex models [51] to quantify the protection factors of face masks and optimize their production and use toward a lower risk in airborne pathogen infection.

### Study Limitations

Some limitations of this study should be considered. First, the 3% accuracy of the flowmeter we used resulted in non-negligible uncertainties on the *F_leak_*, *F_mask_*, and *TFE* derived values, limiting the significance of our results. This aspect can be improved by using measurement instrumentation having higher accuracy. On the other hand, the variability associated with repeated measurements showed a satisfactory repeatability of the test protocol on all tested mask models. Variability between masks from the same production lot were evidenced in literature [52,53], impacting on DP measurement repeatability and possibly on leak quantification.

Second, while the head form used in the study was properly reproducing a standardized and replicable head, it was, however, made of a rigid material with a smooth surface. These features did not perfectly mimic real human skin, which can better adapt to the mask shape, helping to obtain a better face fit and face seal [54]. Therefore, the experimental conditions of this study represented a worst-case scenario, and the resulting leakage data could be overestimated. In particular, our *TFE* results based on BFE measurements reproduced well the filtration efficiency of medium-sized aerosol (3.0 ± 0.3 µm) but possibly underestimated the *TFE* of large droplets generated during speech (>10 µm) [55] and large particles that are more subjected to inertial impaction filtration rather than interception mechanisms [16]. Nonetheless, the relative comparison of leak performance obtained testing different mask models or the same mask model with different wearing styles was sufficiently robust. Further, this study considered a medium sized head form, but the presented methodology could be easily applied to larger and smaller head sizes.

An additional point of weakness is the fact that our study considered only outward flow results. Although previous studies showed that inward and outward protection efficiency are similar for many masks, they can diverge when the mask is worn more loosely or more tightly [56]. Therefore, the outward leak results of this study cannot be directly extended to inward air flow and dedicated testing protocols and equipment should be designed to this end.

Finally, the *TFE* values reported in this study were obtained considering BFE according to EN 14683. BFE tests makes use of bacterial aerosol having a 3.0 ± 0.3 µm mean particle size. An Andersen impactor composed of six stages is used to collect aerosol with aerodynamic sizes ranging from 7.00 to 0.65 μm. The results of *TFE* for smaller aerosol generated during breathing (~1 µm) could be better estimated using the results of particle filtration efficiency (PFE) tests, typically working with 0.3 μm sized particles [41].

## 5. Conclusions

This study provided a simple method for quantifying the fraction of exhaled air leaking from a surgical or community face mask. Results evidenced the key role of filter breathability in guaranteeing a high total filtration efficiency of the face mask. Although an efficient bacterial filtration is a prerequisite to reduce the transmission of airborne respiratory diseases, the presence of a non-negligible fraction of air leaking at the face seal may constitute the main driver impacting on the total filtration efficiency of the mask. Minor changes to filtration performance could be also related to the correct mask wearing. Based on these results, we recommend selecting highly breathable filtering materials for the production and use of face masks, thus minimizing air leaks and maximize user comfort. It is also advised to follow the instructions for use to correctly wear the face mask, and to adjust the nose piece shape and the lace tension to optimize face fit and achieve a balance between user comfort and mask filtration efficiency. An efficient mask, worn well, holds primary importance to curb disease transmission during the COVID-19 pandemic.

## Figures and Tables

**Figure 1 ijerph-19-03548-f001:**
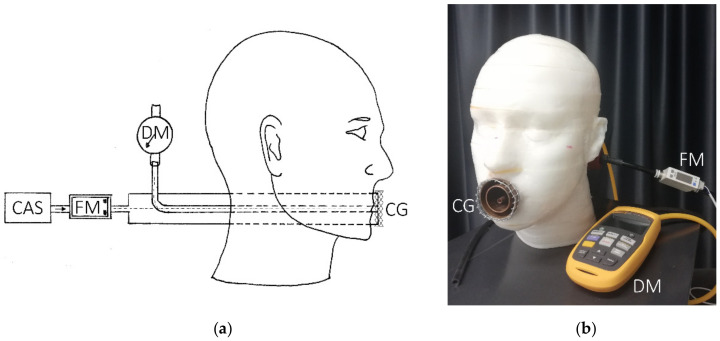
(**a**) Schematic representation of the experimental setup showing the dummy head instrumented with the pipe system, the outlet circular grid (CG), the connected differential manometer (DM), and the compressed air supply (CAS) equipped with the flow meter (FM); (**b**) picture of the experimental setup used in the study.

**Figure 2 ijerph-19-03548-f002:**
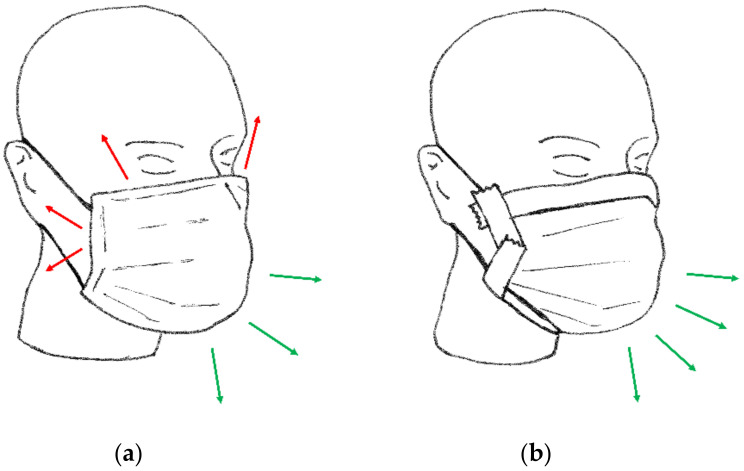
Schematic representation of the two phases of the experimental protocol: (**a**) Phase *I*, (**b**) Phase *II*. The airflow passing through the leaks (red arrows) and through the mask (green arrows) is indicated qualitatively.

**Figure 3 ijerph-19-03548-f003:**
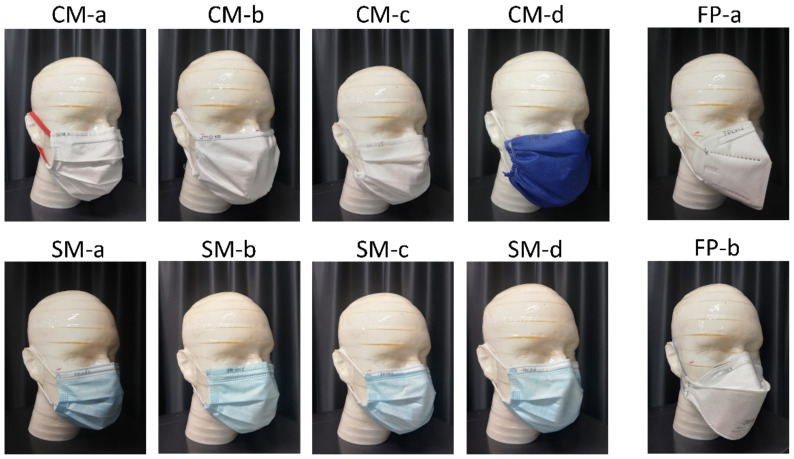
Face masks tested in this study, mounted on the dummy head according to instructions for use. The face mask models included four community masks (CM) and four surgical masks (SM). Two face piece respirator (FP) models were also considered for their mask design specifically conceived to minimize the leakage.

**Figure 4 ijerph-19-03548-f004:**
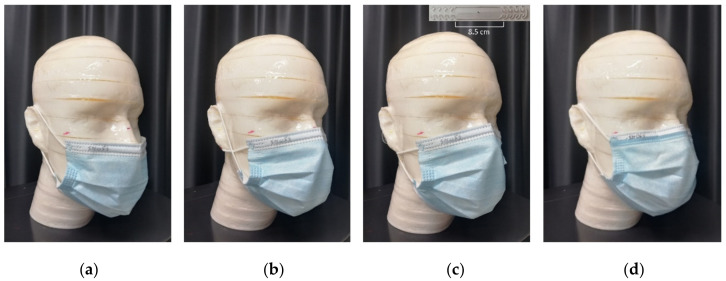
The three mask wearing styles tested in the study (**a**–**c**) alternative to the typical wearing according to instructions for use (IFU) (**d**). (**a**) Mask lowered with the upper band of the mask on nose tip (NOSE), (**b**) mask with laces crossed at the ears (EAR), and (**c**) mask with laces gathered at the nape using an ear saver (see inset) set at a distance of 8.5 cm (NAPE).

**Figure 5 ijerph-19-03548-f005:**
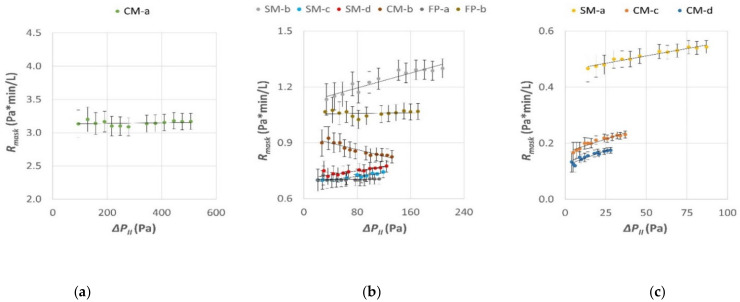
Resistance of mask filter (*R_mask_*) as a function of the differential pressure (Δ*P_II_*) between inner and outer mask filter sides. Dots represent resistance values calculated from experimental data. The first-order polynomial curves that best fitted each mask dataset are also indicated. Data are presented in separate panels (**a**–**c**) using different *x* and *y*-axis ranges for better visualization.

**Figure 6 ijerph-19-03548-f006:**
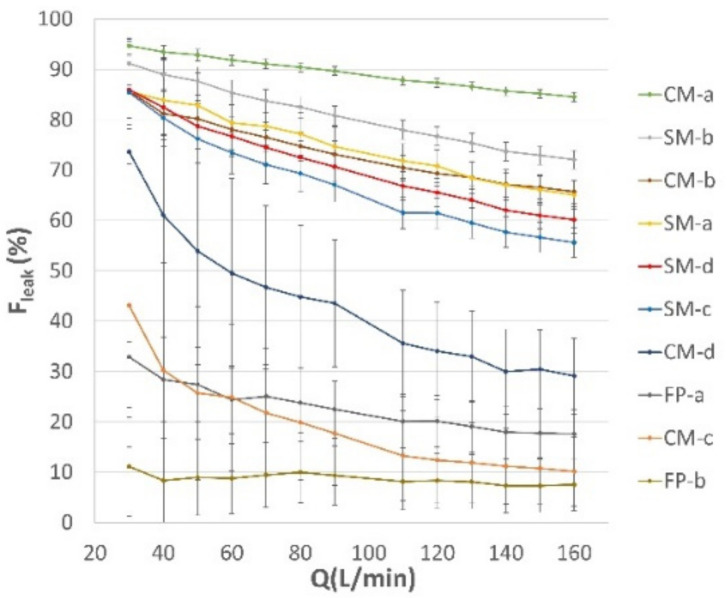
Percentage of airflow passing through the leaks (leak fraction, *F_leak_*) as a function of the total airflow rate. Data are reported for the tested surgical masks (SM), community masks (CM) and face piece (FP) respirators when worn according to manufacturer’s instructions for use.

**Figure 7 ijerph-19-03548-f007:**
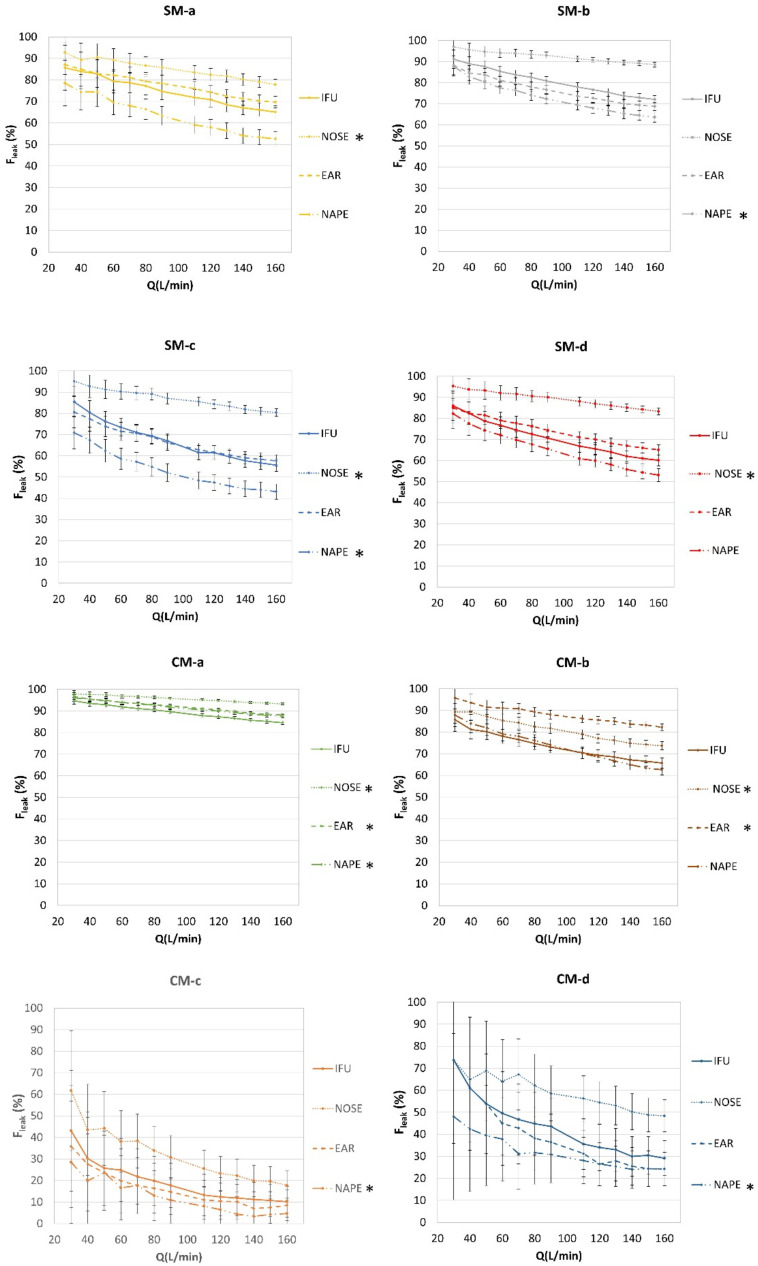
Leakage fraction of the tested surgical masks (SM) and community masks (CM) when worn according to different wearing styles: manufacturer’s instructions for use (IFU); upper band of the mask on the nose tip (NOSE); laces crossed at the ears (EAR); laces gathered at the nape (NAPE). Data are reported for a total outward airflow ranging from 30 L/min to 160 L/min. * *p* < 0.05 indicates significant difference in respect to IFU.

**Figure 8 ijerph-19-03548-f008:**
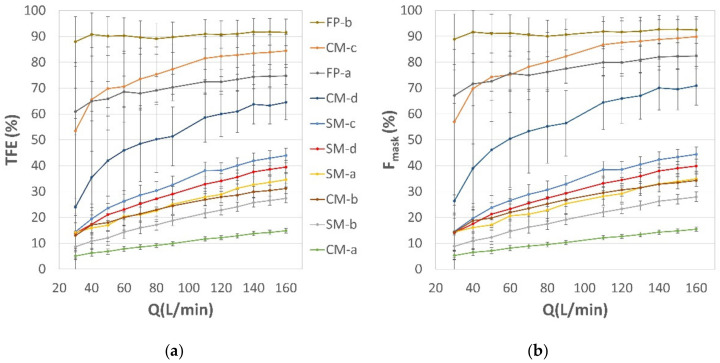
(**a**) Total filtration efficiency (*TFE*) and (**b**) mask airflow fraction (*F_mask_*) of the tested surgical masks (SM), community masks (CM) and face piece (FP) respirators. The comparison of corresponding curves in the two graphs evidences the major role of *F_mask_* in determining TFE, which is affected only in part by BFE values (see, for example, the changes in ranking of SM-c and SM-d between the two graphs).

**Table 1 ijerph-19-03548-t001:** Main specifications and characteristics of the community masks (CM) and surgical masks (SM) tested in this study. Details of the two considered face piece (FP) respirators are also reported.

Mask ID	DP (Pa/cm^2^)	BFE (%)	Mask Type (Manufacturer Claim)	Filter Material	No. Layers (Details)	Fiber Structure	Total Mask Size (cm^2^)	Filtering Area (cm^2^)	Fitting System	Nose Piece
CM-a	315	96	Community, reusable	100% cotton	2	Woven fabric	393	329	Ear loops	Metal wire
CM-b	56	91	Community, reusable	92% cotton 8% PU	2	Knitted fabric	259	225	Ear loops	None
CM-c	10	94	Community, reusable	PP	3 (SSS)	Non-woven	347	308	Ear loops	None
CM-d	7	91	Community, single use	PP	1 (S)	Non-woven	396	356	Ear loops	None
SM-a	28	>99	Surgical Type I, single use	PP	3 (SMS)	Non-woven	271	188	Ear loops	Metal wire
SM-b	77	98	Surgical Type I, single use	PP	3 (SSS)	Non-woven	286	207	Ear loops	Metal wire
SM-c	35	>99	Surgical type I, single use	PP	3 (SMS)	Non-woven	277	200	Ear loops	Metal wire
SM-d	30	>99	Surgical Type IIR, single use	PP	3 (SMS)	Non-woven	272	200	Ear loops	Metal wire
FP-a	58	91	FFP2 respirator	PP	3 (SMS)	Non-woven	252	192	Ear loops	Metal wire
FP-b	53	>99	FFP2 respirator	PP	3 (SMS)	Non-woven	255	173	Head loops	Metal wire + foam

Table abbreviations: DP: breathability according to EN 14683; BFE: bacterial filtration efficiency according to EN 14,683 measured using a bacterial aerosol having a mean particle size of 3.0 ± 0.3 µm; PU: polyurethane; PP: polypropylene; S: spunbonded; M: meltblown.

**Table 2 ijerph-19-03548-t002:** Leak fraction (*F_lea_*_k_) and total filtration efficiency (*TFE*) of tested surgical masks (SM) and community masks (CM) when worn according to different wearing styles: according to manufacturer’s instructions for use (IFU); upper band of the mask on the nose tip (NOSE); with laces crossed at the ears (EAR); with laces gathered at the nape (NAPE). Results obtained for face piece (FP) respirators are indicated only when worn according to IFU. Data are reported for low (30 L/min), medium (90 L/min), and high (160 L/min) outward flowrates.

Mask ID	Wearing Style	*F_leak_* (%) (Mean ± Uncertainty)			TFE (%) (Mean ± Uncertainty)		
		@30 L/min	@90 L/min	@160 L/min	@30 L/min	@90 L/min	@160 L/min
SM-a	IFU	86 ± 10	75 ± 4	65 ± 3	14 ± 10	25 ± 4	35 ± 3
	Nose	93 ± 10	86 ± 4	78 ± 2	7 ± 10	14 ± 4	22 ± 2
	Ear	87 ± 10	78 ± 4	70 ± 3	13 ± 10	21 ± 4	30 ± 3
	Nape	79 ± 11	63 ± 4	53 ± 3	21 ± 11	36 ± 4	47 ± 3
SM-b	IFU	91 ± 4	81 ± 2	72 ± 2	9 ± 4	19 ± 2	27 ± 2
	Nose	97 ± 4	93 ± 2	89 ± 1	3 ± 4	7 ± 2	11 ± 1
	Ear	88 ± 4	77 ± 2	69 ± 2	12 ± 4	23 ± 2	31 ± 2
	Nape	88 ± 4	72 ± 2	64 ± 2	12 ± 4	27 ± 2	36 ± 2
SM-c	IFU	85 ± 7	67 ± 3	56 ± 3	14 ± 7	33 ± 3	44 ± 3
	Nose	95 ± 7	87 ± 3	80 ± 2	5 ± 7	13 ± 3	19 ± 2
	Ear	81 ± 7	66 ± 3	58 ± 3	19 ± 7	34 ± 3	42 ± 3
	Nape	71 ± 8	52 ± 4	43 ± 4	29 ± 8	47 ± 4	56 ± 4
SM-d	IFU	86 ± 7	71 ± 3	60 ± 3	14 ± 7	29 ± 3	39 ± 3
	Nose	95 ± 7	90 ± 2	83 ± 2	5 ± 7	10 ± 2	17 ± 2
	Ear	85 ± 7	7 ± 3	65 ± 2	15 ± 7	25 ± 3	35 ± 2
	Nape	82 ± 7	65 ± 3	53 ± 3	18 ± 7	34 ± 3	46 ± 3
CM-a	IFU	95 ± 2	90 ± 1	85 ± 1	5 ± 2	10 ± 1	15 ± 1
	Nose	98 ± 2	96 ± 1	93 ± 1	2 ± 1	4 ± 1	7 ± 0
	Ear	97 ± 2	92 ± 1	88 ± 1	3 ± 2	7 ± 1	12 ± 1
	Nape	96 ± 2	92 ± 1	88 ± 1	4 ± 2	8 ± 1	12 ± 1
CM-b	IFU	86 ± 5	73 ± 3	66 ± 2	13 ± 5	24 ± 2	31 ± 2
	Nose	89 ± 5	82 ± 2	74 ± 2	10 ± 5	17 ± 2	24 ± 2
	Ear	96 ± 5	88 ± 2	82 ± 2	4 ± 5	11 ± 2	16 ± 1
	Nape	88 ± 5	74 ± 3	63 ± 2	11 ± 5	24 ± 2	34 ± 2
CM-c	IFU	43 ± 28	18 ± 10	10 ± 7	53 ± 26	77 ± 10	84 ± 7
	Nose	62 ± 28	31 ± 10	18 ± 7	36 ± 26	65 ± 9	77 ± 6
	Ear	36 ± 28	15 ± 11	8 ± 7	60 ± 27	80 ± 10	86 ± 7
	Nape	29 ± 28	11 ± 11	5 ± 7	67 ± 27	83 ± 10	89 ± 7
CM-d	IFU	74 ± 38	44 ± 13	29 ± 8	24 ± 34	51 ± 11	64 ± 7
	Nose	74 ± 38	59 ± 13	48 ± 7	24 ± 34	38 ± 11	47 ± 7
	Ear	74 ± 38	36 ± 13	24 ± 8	24 ± 34	58 ± 12	69 ± 7
	Nape	48 ± 38	31 ± 13	24 ± 8	47 ± 34	63 ± 12	69 ± 7
FP-a	IFU	33 ± 10	22 ± 6	18 ± 5	61 ± 9	71 ± 5	74 ± 4
FP-b	IFU	11 ± 10	9 ± 6	8 ± 5	88 ± 10	90 ± 6	91 ± 5

Table abbreviations: *F_leak_*: leak fraction; *TFE*: total filtration efficiency; SM: surgical mask; CM: community masks; IFU: according to manufacturer’s instructions for use; NOSE: with upper band of the mask on the nose tip; EAR: with laces crossed at the ears; NAPE: with laces gathered at the nape.

**Table 3 ijerph-19-03548-t003:** Summary of changes in the leakage fraction (*F_leak_*) of tested surgical masks (SM) and community masks (CM) when worn according to wearing styles alternative to that reported in the instructions for use (IFU): upper band of the mask on the nose tip (NOSE); laces crossed at the ears (EAR); laces gathered at the nape (NAPE).

Mask ID	Changes in *F_leak_* for Wearing Styles Alternative to Instructions for Use		
	IFU vs. NOSE	IFU vs. EAR	IFU vs. NAPE
CM-a	↑ *	↑ *	↓ *
CM-b	↑ *	↑ *	↔
CM-c	↑	↓	↓ *
CM-d	↑	↓	↓ *
SM-a	↑ *	↑	↓
SM-b	↑	↓	↓ *
SM-c	↑ *	↔	↓ *
SM-d	↑ *	↑	↓

Legend: *: *p* < 0.05; ↑ = more leaks compared to IFU is color-coded in red; ↓ = less leaks compared to IFU is color-coded in green; ↔ = same leak amount as IFU (within 5%). Arrows report a qualitative comparison, indicating a higher, lower, or similar amount of F_leak_ over the whole exhaled flow rate (30–160 L/min). Results are color-coded using red or green for wearing styles that worsen or improve the mask leakage performance, respectively. Comparisons showing statistical significance are indicated by *.

## Data Availability

The data presented in this study are available on reasonable request from the corresponding author.

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
