# Peer review of "A Simple Method to Quantify Outward Leakage of Medical Face Masks and Barrier Face Coverings: Implication for the Overall Filtration Efficiency"

_ijerph, 2022, doi:10.3390/ijerph19063548_

Round 1
Reviewer 1 Report
I have read the study of Chiera and colleagues with keen interest and in detail. The study has been well designed and conducted and the findings has been reported in scholarly way. As the authors might know, facemasks (FMs) or particulate-filtering respirators (PFRs) is recommended as a personal-level intervention to reduce exposure to ambient particulate matter air pollution and its health effects by the public-health bodies such as the World Health Organization (WHO), as well as the American Heart Association and the European Society of Cardiology. As a result, the authors need to discuss on this issue in the introduction section. Please see the following papers:
Cardiovascular health effects of wearing a particulate-filtering respirator to reduce particulate matter exposure: A randomized crossover trial
Can respirator face masks in a developing country reduce exposure to ambient particulate matter?
Additionally, the authors need to report the main reasons of low efficiency of FMs or PFRs against ambient particulate matter air pollution in the discussion section.
Author Response
Ref.: Ms. No. ijerph-1617578
Response to Reviewer # 1
We like to thank the reviewer for the careful evaluation of our manuscript and for the useful comment, which have helped improving our manuscript.
Changes to manuscript text are highlighted in yellow.
Below we respond point by point to the various comments and indicate the changes we made in the revised manuscript.
Q11: I have read the study of Chiera and colleagues with keen interest and in detail. The study has been well designed and conducted and the findings has been reported in scholarly way. As the authors might know, facemasks (FMs) or particulate-filtering respirators (PFRs) is recommended as a personal-level intervention to reduce exposure to ambient particulate matter air pollution and its health effects by the public-health bodies such as the World Health Organization (WHO), as well as the American Heart Association and the European Society of Cardiology. As a result, the authors need to discuss on this issue in the introduction section. Please see the following papers:
Cardiovascular health effects of wearing a particulate-filtering respirator to reduce particulate matter exposure: A randomized crossover trial
Can respirator face masks in a developing country reduce exposure to ambient particulate matter? Additionally, the authors need to report the main reasons of low efficiency of FMs or PFRs against ambient particulate matter air pollution in the discussion section.
A11: We thank the reviewer for rising this point about effects of wearing facemasks and particle filtering respirators on the cardiovascular system. In order to avoid further lengthening of the manuscript introduction, we shortly mentioned this aspect in the introduction section (lines 41-42) and took the opportunity to address particle filtration efficiency (PFE) in the discussion section (lines 574-579). A couple of references were also added including those suggested by the reviewer.
Reviewer 2 Report
The research is timely, novel, and needed. The manuscript was well prepared and written in an organized and clear manner. A few suggestions and comments are provided in the attached document.
One point to consider is the discussion of why the research did not consider evaluating the particle filtration efficiency (PFE) values at the 0.3 micron level to provide insights and comparisons to inward leakage/respiratory protection. I do not think the 0.3 micron values need to be provide for the manuscript to be published, as I believe their is value in the BFE comparison, however, a brief acknowledgement and discussion of the PFE testing would be beneficial.

Author Response
Ref.: Ms. No. ijerph-1617578
Response to Reviewer # 2
We like to thank the reviewer for the positive evaluation of our manuscript and for the useful comments, which have helped improving our manuscript.
Changes to manuscript text are highlighted in yellow.
Below we respond point by point to the various comments and indicate the changes we made in the revised manuscript.
Q21: The research is timely, novel, and needed. The manuscript was well prepared and written in an organized and clear manner. A few suggestions and comments are provided in the attached document.
One point to consider is the discussion of why the research did not consider evaluating the particle filtration efficiency (PFE) values at the 0.3 micron level to provide insights and comparisons to inward leakage/respiratory protection. I do not think the 0.3 micron values need to be provide for the manuscript to be published, as I believe there is value in the BFE comparison, however, a brief acknowledgement and discussion of the PFE testing would be beneficial.
A21: We thank the reviewer for the positive evaluation of the manuscript and the valuable comments. We fully agree with the reviewer about the complementary information that could be obtained from PFE test other than BFE test. Indeed, the two testing methodologies provide information about different dimensional ranges. Our model of flow repartition does not consider the effect of the momentum of the particles, thus being insensitive to particle size. However, particles with low momentum (i.e. size) are more prone to follow the airflow, therefore TFE data are more accurate to describe these particles pathways than the pathway of those with high momentum that are more prone to be collected by impaction mechanisms on the filter. These aspects were addressed in more details in the discussion of the revised manuscript (lines 560-579).
Q22: There is a push to refer to these products as barrier face coverings now that the ASTM F3502 standard is in use. Changing to this terminology might help to alleviate any confusion.
A22: We agree with the reviewer and updated the manuscript title to include this terminology. Moreover, we added the term “barrier face coverings” among the terminologies described in the Introduction section (lines 57-59).
Q23: The ASTM F3502 Standard Specification for Barrier Face Coverings was adopted in Feb. 2021. It doesn't regulate the performance but does offer a framework for the specification of performance. Some discussion of this standard specification would be good to include in this section.
A23: We mentioned the ASTM F3502 in the introduction section as the first standard attempting to define testing methods of non-medical face masks (lines 66-69).
Q24: Was there any thought to including the particulate filtration efficiency at the lower particle size instead of the BFE test?
A24: Of course the idea of including PFE performance in addition to BFE was considered by the authors. Unfortunately, PFE data of tested masks were not available. However, it is worth noting that the flow repartition data we obtained according to our methodology makes no assumption on the presence of particulate or aerosol within the exhaled air. Therefore, the proposed TFE parameter can be recalculated by using PFE instead of BFE. Reliable TFE results could be obtained whenever the momentum of the particles is low enough to guarantee that a negligible fraction of the particles is collected on the mask filter by inertial impaction. This point has been addressed in the discussion of the revised manuscript (lines 574-579).
Q25: You could add a note to explicitly state the particle size for the BFE for the reader that may not be familiar with the test.
A25: A note to Table 1 indicating the size of the aerosol generated in BFE test has been added as suggested by the reviewer (line 265). This information is also reported in the discussion section for sake of clarity and to better compare BFE with PFE typical particle size (line 575).
Q26: It is a little difficult to figure out which line goes with which label. Either add the label to each line on the graph or put the legend in an order close to the way the lines appear on the graph. You could also use different marker shapes to better identify each line.
A26: We thanks the reviewer for his/her suggestion on how to improve the readability of Figures 6 and 8. We chose to put the legend in an order close to the way the lines appear on the graph.
Q27: The "leak" needs to be subscripted so that it doesn't read as "Fleak".
A27: Amended as suggested.
Reviewer 3 Report
This paper presents a novel method of measurement of bypass leakage in face masks. This helps to fill a gap in measurement techniques which currently rely human volunteers and expensive particle counting methods. The methods are sound and the conclusions are well-justified and important for producers of masks and regulatory bodies. The paper is generally well structured and written.
General points
Introduction and conclusions:
Covid is important but this work is relevant to all respiratory infections and other airborne particulate hazards, not just covid, and masks are likely to to be applied to these forever especially in healthcare settings.
All figures but especially Figures 5 and 7 - graph bitmaps are difficult to read due to low resolution and overcompression. Please correct for final version.
Specific points
34: Most would agree that omicron is transmitted mostly from the upper airway due to the high viral load there, however this is not clear for delta and earlier variants with higher viral loads in the lower airway, especially in severe cases. It is a complex issue which can be avoided by referring to airway only, without specifying upper or lower.
71- There is a sentence missing about user fit testing as used with EN 149 masks. The tests can measure inward leakage in situ but typically with sub-micron aerosol, which EN 14683 and community masks are not designed to filter.
112 & 134 Ed. subscript not footer
177- For mass conservation these flows should be corrected to standard conditions - the flow meter probably does this automatically but even if it doesn't the differences in absolute pressure are pretty small (<0.5%). Would be helpful to state whether the flows are ambient or corrected.
180 - Please specify type of adhesive tape, and any procedure used to apply it and check the seal was made. It can be difficult to achieve a good seal with some tapes and fabrics. Was this a source of the large uncertainties seen for some masks?
282 Ed.- except instead of but
288- Do I understand correctly that a "lift off" effect is not seen, where bypass increases with higher flow? This is a useful point to note if correct.
342 -Ed. some words jumbled - should this read "CM-a had very poor breathability"?
358- Comment on whether FP-a would be compliant with EN149 if tested accordingly (as was mentioned for SM-b above)- it seems unlikely that it could with such a poor BFE and high bypass leakage.
455- ed. propensity not propension
532- An additional limitation to mention is that only one size of head was used- typically crossing earloops and attaching earloops behind the neck is recommended for smaller heads.
532 A point is missing here that BFE does not represent aerosol from breath ~1µm nor aerosol from speech >10µm. For breath aerosol the bypass would be worse, and for speech aerosol the bypass wouldn't be as bad. Up to the authors whether to include this point. Lots of references for breath aerosol e.g.
classical:
Johnson, G. R., L. Morawska, Z. D. Ristovski, M. Hargreaves, K. Mengersen, C. Y. H. Chao, M. P. Wan, et al. ‘Modality of Human Expired Aerosol Size Distributions’. Journal of Aerosol Science 42, no. 12 (1 December 2011): 839–51. https://doi.org/10.1016/j.jaerosci.2011.07.009.
or more recent:
Bagheri, Gholamhossein, Oliver Schlenczek, Laura Turco, Birte Thiede, Katja Stieger, Jana-Michelle Kosub, Mira L. Pöhlker, et al. ‘Exhaled Particles from Nanometre to Millimetre and Their Origin in the Human Respiratory Tract’, 3 October 2021. https://doi.org/10.1101/2021.10.01.21264333.
Author Response
Ref.: Ms. No. MEAS-D-21-04410
Response to Reviewer # 3
We like to thank the reviewer for the careful evaluation of our manuscript and for the useful comments, which have helped improving our manuscript.
Changes to manuscript text are highlighted in yellow.
Below we respond point by point to the various comments and indicate the changes we made in the revised manuscript.
Q31: Introduction and conclusions: Covid is important but this work is relevant to all respiratory infections and other airborne particulate hazards, not just covid, and masks are likely to be applied to these forever especially in healthcare settings.
A31: We thank the reviewer for suggesting to briefly address the impact of face masking, and other infection prevention strategies, on several infectious respiratory diseases in the introduction section. This aspect has been mentioned in the revised abstract, and a sentence and two references were added in the introduction section (lines 41-42).
Q32: All figures but especially Figures 5 and 7 - graph bitmaps are difficult to read due to low resolution and overcompression. Please correct for final version.
A32: We apologize for the low resolution of the figures in the pdf draft. Actually they were uploaded on the editorial manager in high resolution according to the Journal requirements and are part of the material sent to the editor.
Q33: Line 34-Most would agree that omicron is transmitted mostly from the upper airway due to the high viral load there, however this is not clear for delta and earlier variants with higher viral loads in the lower airway, especially in severe cases. It is a complex issue which can be avoided by referring to airway only, without specifying upper or lower.
A33: We agree with the reviewer. The reference to the sole upper airways was removed.
Q34: Line71 - There is a sentence missing about user fit testing as used with EN 149 masks. The tests can measure inward leakage in situ but typically with sub-micron aerosol, which EN 14683 and community masks are not designed to filter.
A34: The reference to fit testing using according to EN 149 specifications is now indicated in the revised manuscript version, specifically referencing EN 149 which include the test for inward leakage in situ using sub-micron aerosol made of sodium chloride and paraffin oil as specified in EN 13274-7.
Q35: Lines 112 & 134 - Ed. subscript not footer.
A35: Amended as indicated
Q36: Line 177 - For mass conservation these flows should be corrected to standard conditions - the flow meter probably does this automatically but even if it doesn't the differences in absolute pressure are pretty small (<0.5%). Would be helpful to state whether the flows are ambient or corrected.
A36: The flow meter used in this study provided volumetric flow measurement corrected to standard conditions (101.3 kPa, 20°C, 65% RH). This information has been added to the material and methods section (lines 165-166).
Q37: Line 180 - Please specify type of adhesive tape, and any procedure used to apply it and check the seal was made. It can be difficult to achieve a good seal with some tapes and fabrics. Was this a source of the large uncertainties seen for some masks?
A37: The tape used in the study was a paper masking tape, 25 mm width (Tesa® Masking Economy, Tesa SE Hamburg, Germany) as now detailed in the materials and methods section of the revised manuscript (lines 191-198). The masking tape was applied across the mask filter perimeter and the surface of the dummy head. Special care was paid to cover only the peripheral welded areas of the mask filter that do not contribute to the airflow through the mask. Manual pressure was made on the tape to guarantee no air leaking was present though the sealed areas. A visual inspection of the tape adherence to the mask boundaries and to the dummy head was performed at the end of each Phase II experiment. This procedure allowed to assume that zero leakage condition was satisfied.
According to the results obtained in preliminary experiments (data not reported in the manuscript), we obtained good measurement repeatability, thus supporting the achievement of an effective sealing in different experimental sessions. Moreover, we tested several times the tape sealing efficacy by comparing differential pressure measurements performed at a flow rate of 30 l/min in Phase II experimental conditions before and after increasing the flow rate to 160 l/min.
The magnitude of the large uncertainties associated to some of the tested CMs and SMs was mainly related to the instrumental accuracy (3% + 1LSD) of the flowmeter combined with the instrumental accuracy (1% + 1LSD) of the differential manometer. The combination of instrumental uncertainties was significantly impacting the values of Fleak at low flow rates for masks with low DP. This point is part of the study limitations.
Q38: Line 282 Ed.- except instead of but
A38: Amended as suggested
Q39: Line 288- Do I understand correctly that a "lift off" effect is not seen, where bypass increases with higher flow? This is a useful point to note if correct.
A39: We thank the reviewer for properly suggesting this additional reason for the atypical behavior of CM-b mask resistance at higher differential pressure. Indeed, a “lift off" effect in the area close to the nose of the dummy head could have occurred due to the absence of a nose piece and the elasticity of the mask boundary. This additional point was addressed in the fist section of the Results (lines 309-312).
Q310: Line 342 -Ed. some words jumbled - should this read "CM-a had very poor breathability"?
A310: Amended as suggested
Q311: Line 358- Comment on whether FP-a would be compliant with EN149 if tested accordingly (as was mentioned for SM-b above)- it seems unlikely that it could with such a poor BFE and high bypass leakage.
A311: We agree with the reviewer. The FP-a respirator tested in this study is likely to fail the inward leakage requirements set in the EN 149 standard, despite it was labeled as EN 149 compliant by the manufacturer. However, a direct comparison between outward and inward leakage values is not straightforward (e.g. possible lift-off effects could be present, especially in FFP respirators without exhalation valves). Taking this into consideration, we added a new sentence in the revised manuscript commenting on this aspect (lines 373-377).
Q312: 455- ed. propensity not propension.
A312: Amended as suggested
Q313: Line 532- An additional limitation to mention is that only one size of head was used- typically crossing earloops and attaching earloops behind the neck is recommended for smaller heads.
A313: As suggested by the reviewer, the importance of considering variability in head size has been added to the revised Discussion section (lines 566-567).
Q314: 532 A point is missing here that BFE does not represent aerosol from breath ~1µm nor aerosol from speech >10µm. For breath aerosol the bypass would be worse, and for speech aerosol the bypass wouldn't be as bad. Up to the authors whether to include this point. Lots of references for breath aerosol e.g.
classical:
Johnson, G. R., L. Morawska, Z. D. Ristovski, M. Hargreaves, K. Mengersen, C. Y. H. Chao, M. P. Wan, et al. ‘Modality of Human Expired Aerosol Size Distributions’. Journal of Aerosol Science 42, no. 12 (1 December 2011): 839–51. https://doi.org/10.1016/j.jaerosci.2011.07.009.
or more recent:
Bagheri, Gholamhossein, Oliver Schlenczek, Laura Turco, Birte Thiede, Katja Stieger, Jana-Michelle Kosub, Mira L. Pöhlker, et al. ‘Exhaled Particles from Nanometre to Millimetre and Their Origin in the Human Respiratory Tract’, 3 October 2021. https://doi.org/10.1101/2021.10.01.21264333.
A314: The importance of aerosol and particle size in the definition of total filtration efficiency has been further addressed in the Study limitations section of the revised manuscript (lines 560-563). The role of inertial impaction filtration mechanisms has been pointed out for particles generated during speech. For particles smaller than those generated during BFE testing, we indicated the possibility to complement TFE data based on BFE with those based on filter PFE (lines 574-579).